# Extreme Parkour with Legged Robots

**Anonymous**

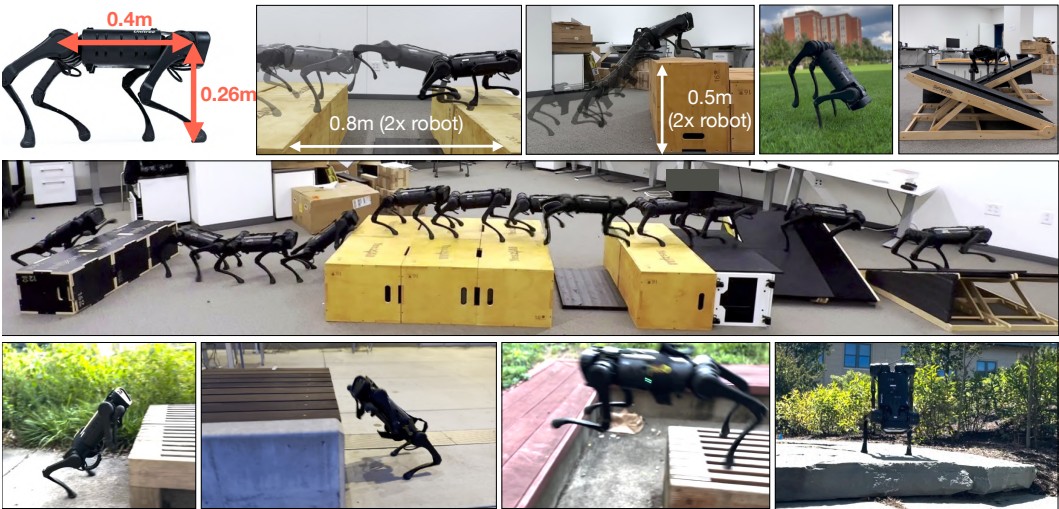

Figure 1: **Extreme Parkour**: Low-cost robot with imprecise actuation can perform precise athletic behaviors directly from a high-dimensional image without any explicit mapping and planning. The robot is able to long jump across gaps $2\times$ of its own length, high jump over obstacles $2\times$ its own height, run over tilted ramps, and walk on just front two legs (handstand) – all with *a single neural network operating directly on depth from a single, front-facing camera.*

**Abstract:** Humans can perform parkour by traversing obstacles in a highly dynamic fashion requiring precise eye-muscle coordination and movement. Getting robots to do the same task requires overcoming similar challenges. Classically, this is done by independently engineering perception, actuation, and control systems to very low tolerances. This restricts them to tightly controlled settings such as a predetermined obstacle course in labs. In contrast, humans are able to learn parkour through practice without significantly changing their underlying biology. In this paper, we take a similar approach to developing robot parkour on a small low-cost robot with imprecise actuation and a single front-facing depth camera for perception which is low-frequency, jittery, and prone to artifacts. We show how a single neural net policy operating directly from a camera image, trained in simulation with large-scale RL, can overcome imprecise sensing and actuation to output highly precise control behavior end-to-end. We show our robot can perform a high jump on obstacles 2x its height, long jump across gaps 2x its length, do a handstand and run across tilted ramps, and generalize to novel obstacle courses with different physical properties.

7th Conference on Robot Learning (CoRL 2023), Atlanta, USA.

# 1  Introduction

Parkour is a popular athletic sport that involves humans traversing obstacles in a highly dynamic manner like running on walls and ramps, long coordinated jumps, and high jumps across obstacles. This involves remarkable eye-muscle coordination since missing a step can be fatal. Further, because of the large torques exerted, human muscles tend to operate at the limits of their ability and limbs must be positioned in such a way as to maximize mechanical advantage. Hence, margins for error are razor thin, and to execute a successful maneuver, the athlete needs to make all the right moves. Understandably, this is a much more challenging task than walking or running and requires years of practice to master. Replicating this ability in robotics poses a massive software as well as hardware challenge as the robot would need to operate at the limits of hardware for extreme parkour. Perception and control must be precise and tightly coupled to execute the correct moves at the right time. The robot should have a precise physical understanding and be able to come up with correct moves on the fly in advance of the obstacle because, unlike locomotion, recovering from suboptimal behavior is not only unsafe but also makes it impossible to do the task. For instance, jumping over a wide gap needs enough time to generate the required momentum to take off before the edge. Hence, classical approaches can only do parkour when everything is pre-measured precisely, that is, placement, size, and type of obstacle course are known and optimization is performed to decide the right control actions at each timestep [1]. But what if any obstacle were to move, or if the robot is asked to perform as is on a new parkour course? All these challenges are not feasible with such an approach.

In contrast, humans take a very different approach. A parkour expert and novice have access to the same set of "sensors", and in the process of learning parkour, their sensing capabilities are not significantly improved. Instead, through years of trial and error, they learn to use the same imprecise sensing and actuation to accomplish amazing feats in in-the-wild settings. In this paper, our hypothesis is that we can demonstrate learning parkour in a similar way on low-cost robots.

We build upon the recent line of works that show impressive results on walking and running in diverse scenarios [2, 3, 4, 5, 6, 7, 8, 9, 10, 11] and use the low-cost Unitree A1 hardware. However, low cost poses a new challenge for parkour which is not as prominent in prior walking works. Due to the noisy and laggy action, the perception has artifacts, latency, and jitter [6]. Hence, building a terrain map with noisy perception leads to large errors in the map which throws off the action planner. Even if the actions were correct, executing them on laggy and noisy actuators leads to catastrophic failure.

In addition to precise control from noisy actuation, training extreme parkour controllers has two conceptual challenges as well. First, the robot should have the "freedom" to automatically adjust its heading direction depending on the type of parkour obstacle. We found that even if a human expert is providing the heading direction, it is sub-optimal because in extremely long or high jumps over obstacles or ramps, even a few degrees of error in heading leads to failure. Second, each parkour behavior from jumping to handstand are very different in nature, so combining them within a single neural network is a challenging learning problem.

We address all these challenges with an end-to-end data-driven reinforcement learning framework. A single neural network is trained via RL in simulation to directly output motor commands from pixels [2, 3, 7]. To allow the robot to adjust itself as per the obstacle type at deployment, we propose a novel dual distillation method. The policy is first trained with a privileged heading in Phase 1 and then distilled to predict its own heading direction in Phase 2. As a result at deployment, the policy not only outputs agile motor commands but also rapidly adjusts heading directions all from input depth image. Furthermore, to allow a single neural network to represent diverse parkour skill behaviors we propose a simple yet effective universal reward design principle based on inner-products. Below, we summarize the main contributions:

- A novel dual distillation method for distilling both agile motor commands and rapidly fluctuating heading directions from depth images.

- A simple yet effective inner-product reward design principle for general robot base motion acquisition, together with an automatic terrain curriculum for overcoming exploration in RL.

- Our results set a new landmark in learning-driven parkour with high jumps that are **2x** the height of the robot, long jumps that are **2x** the length of the robot, walking on front two legs (handstand), and jumping over titled ramps directly from a single front-facing egocentric depth camera (Tab. 1).

## 2 Related Work

### 2.1 Legged Locomotion

Classical approaches for locomotion use model-based control to define walking controllers [15, 16, 17, 18, 19] and combine them with elevation maps constructed by fusing point cloud and odometry data [20, 21, 22, 23, 24, 25, 26, 27, 28, 29]. However, these controllers can struggle to generalize to situations with widely varying physical properties such as ice or deformable material. This has motivated the use of learned controllers trained with RL that can adapt to changes in dynamics [8, 30, 31, 32, 33, 34, 35] and also leverage elevation maps [6, 36, 37, 38, 39] for perceptive walking. Building elevation maps usually requires sophisticated sensors and causes artifacts that degrade downstream performance. Recent work skips the use of elevation maps entirely and accomplishes highly robust perceptive walking [2, 7, 3, 40]. In this paper, we generalize a similar paradigm with key modifications to parkour.

| Method | Robot | Climb | Gap | Ramp | Handstand |
|---|---|---|---|---|---|
| Rudin et. al [12] | AnymalC | 1.1 | 0.75 | × | × |
| Hoeller et. al [13]* | AnymalC | 2 | 1.5 | × | × |
| Zhuang et. al [14]* | Unitree-A1 | 1.6 | 1.5 | × | × |
| Extreme Parkour (ours) | Unitree-A1 | **2** | **2** | **37°** | ✓ |

Table 1: Comparison of parkour setups. Starred papers in 2nd and 3rd row are concurrent works (recently released). The numbers in Climb and Gap denote the relative size of obstacles with respect to quadruped's height and length respectively. Notably, our method is able to push the low-cost A1 robot to extreme scenarios which are *twice* the height and length of robot. Anymal is an industry-standard high-quality robot and therefore much more expensive.

### 2.2 Robotic Parkour

Most animals and humans learn locomotion within the first year of their life. In contrast, parkour is more challenging and requires years to master since a single error can lead to failure. Results on this task are comparatively fewer although recent years have seen some progress [1]. [12] use the classical approach of decomposing perception into elevation mapping and use RL to train a policy conditioned on it. Some recent work demonstrates blind dynamic running and jumping using sim2real RL on quadrupeds [41] and bipeds [9, 10].

### 2.3 Concurrent work

There are two other concurrent works released within weeks. [13] demonstrates agile behaviors by training task-specific policies and composing them using a high-level trained module but still relying on elevation maps. And [14] trains an end-to-end policy that uses depth instead of elevation map but needs a complex curriculum of first training with soft penetration constraints in simulation followed by distillation to hard constraints. They also use simplified obstacle abstractions (type, width, height, and robot's distance to the obstacle) as privileged visual information. However, this type of information cannot be generalized to general obstacle geometries. In contrast to both of these papers, we propose a conceptually simple framework that results in more extreme parkour behaviors. The simplicity comes from three ideas: (i) instead of privileged abstractions, we use scandots as privileged information that generalizes across terrain geometries, (ii) allowing the policy to decide its own heading at deployment depending on obstacles. This allows us to demonstrate the capability of jumping across tilted ramps. And (iii) a unified general-purpose reward principle. Furthermore, we are able to cross gaps that are upto $2\times$ the length of the robot and jump obstacles that are $2\times$ its height, whereas concurrent work jumps at most $1.5\times$ its height and $1.5\times$ its length (Tab. 1).

## 3 Method

We wish to train a single neural network that goes directly from raw depth and onboard sensing to joint angle commands. To train adaptive motor policies, recent approaches use two-phase student teacher training [8, 6, 42, 43]. Later works [44, 45] introduce regularized online adaptation (ROA) to collapse this into a single phase. To train the vision backbone, a similar teacher-student framework

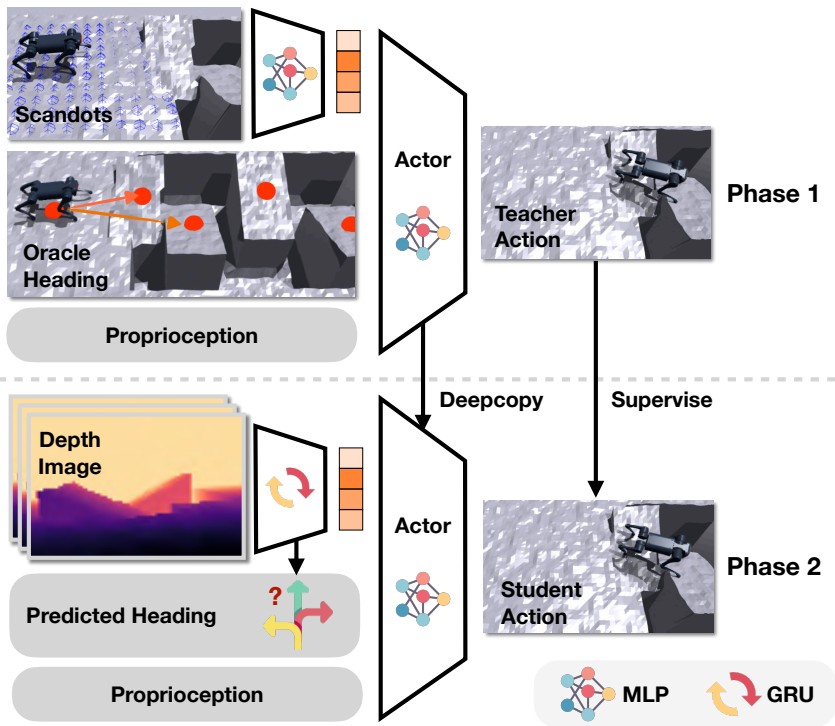

Figure 2: Training overview. In phase 1, we use RL to learn a locomotion policy with access to privileged information like environment parameters and scandots [2] in addition to heading direction from waypoints. We use Regularized Online Adaptation (ROA)[31] to train an estimator to recover environmental information from the history of observations. In phase 2, we distill from scandots into a policy that operates from onboard depth and *automatically decides its heading (yaw) direction* conditioned on the obstacle.

is employed [2, 3, 7] where a teacher trained with privileged scandots information is distilled to a student with access to depth. In this paper, we use ROA for adaptation and two-phase training for the vision backbone but introduce key modifications for the challenging task of extreme parkour.

First, since parkour requires diverse behaviors to traverse different obstacles it is challenging to engineer reward functions specific to each. We present a simple, unified reward formulation from which diverse behaviors emerge automatically and are perfectly adapted to the terrain geometry. Second, during parkour the robot needs to be able to choose its own direction as opposed to following human-specified ones. For instance, when jumping across tilted ramps, it needs to jump on the first ramp at a very specific angle and then change directions immediately which is impossible for a human to provide. Instead, we provide directions in phase 1 using suitably placed waypoints and in phase 2 we train a network to predict these oracle heading directions from depth information.

## 3.1 Unified Reward for Extreme Parkour

The rewards used in [2] do not transfer directly to the parkour case. The robot cannot follow arbitrary direction commands and instead must have the freedom to choose the optimal direction. Instead of randomly sampling directions, we compute direction using waypoints placed on the terrain (Fig. 3) as

$$\hat{\mathbf{d}}_w = \frac{\mathbf{p} - \mathbf{x}}{\|\mathbf{p} - \mathbf{x}\|} \tag{1}$$

where $\mathbf{p}$ is the next waypoint location and $\mathbf{x}$ is robot location in the world frame. The velocity tracking reward is then computed as

$$r_{tracking} = \min(\langle \mathbf{v}, \hat{\mathbf{d}}_w \rangle, v_{cmd}) \tag{2}$$

where $\mathbf{v} \in \mathbb{R}^2$ is the robot's current velocity in world frame and $v_{cmd} \in \mathbb{R}$ is the desired speed. Note that [2] tracks velocity in the base frame but world frame is used. This is done to prevent the robot from exploiting the reward and learning the unintended behavior of turning around the obstacle.

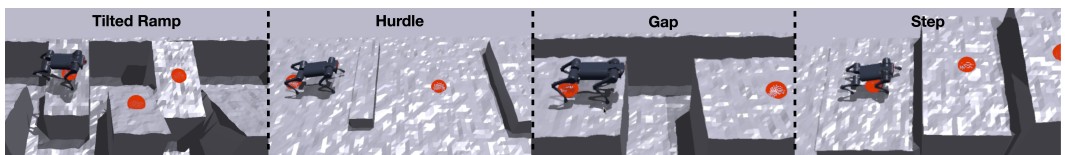

Figure 3: Terrains in simulation with red dots indicating waypoints that are used to get heading direction.

While the above reward is sufficient for diverse parkour behavior, for challenging obstacles the robot tends to step close to the edge to minimize energy usage. This behavior is risky and does not transfer well to real settings. We therefore add a term to penalize foot contacts near terrain edges.

$$r_{clearance} = -\sum_{i=0}^{4} c_i \cdot M[p_i] \tag{3}$$

$c_i$ is 1 if $i$th foot touches the ground. $M$ is a boolean function which is 1 *iff* the point $p_i$ lies within 5cm of an edge. $p_i$ is the foot position for each leg.

The rewards defined above typically lead to a gait that uses all four legs. However, a defining feature of parkour is walking in different styles that are aesthetically pleasing but may not be biomechanically optimal. To explore this diversity, we introduce a term to track a desired forward vector using the same inner product design principle, which can be controlled by the operator at test time.

$$r_{stylized} = W \cdot \left[0.5 \cdot \langle \hat{\mathbf{v}}_{fwd}, \hat{\mathbf{c}} \rangle + 0.5\right]^2 \tag{4}$$

where $\hat{\mathbf{v}}_{fwd}$ is the unit vector pointing forward along the robot's body, $\hat{\mathbf{c}}$ is also a unit vector indicating the desired direction and $W$ is a binary number to switch the reward on/off. In our case, we train the robot to do a handstand and choose $\hat{\mathbf{c}} = [0,0,-1]^T$. $W$ is sampled randomly in $\{0,1\}$ at training and controlled via remote at deployment. We also use the additional regularization terms from [45].

### 3.2 Reinforcement Learning from Scandots (Phase 1)

We use the above rewards to learn a policy using model-free RL [46] in simulation. This policy takes as input, the proprioception $\mathbf{x}$, scandots $\mathbf{m}$, target heading $\hat{\mathbf{d}}$, walking flag $W$ and commanded speed $v^{cmd}$. We use regularized online adaptation (ROA) [44] to train an adaptation module to estimate environment properties. We create a set of tilted ramps, gaps, hurdles and high step terrains (Fig. 3), and arrange them in increasing difficulty as in [2]. To aid exploration, robots are first initialized in easy levels. They are promoted to harder ones if they traverse more than half the length, and demoted to an easier one if they travel less than half the expected distance $v^{cmd}T$ ($T$ is episode length).

### 3.3 Distilling Direction and Exteroception (Phase 2)

The phase 1 policy relies on two pieces of information not directly available on the real robot. First, exteroceptive information is only available in the form of depth images from a front-facing camera instead of scandots. Second, there is no expert to specify waypoints and target directions, these must be inferred from the visible terrain geometry. We use supervised learning to obtain a deployable policy which automatically estimates these quantities. For exteroception, similar to the RMA architecture in [2] we replace the scandots input to the base policy with a convnet-GRU pipeline that accepts depth. This network is trained using DAgger [47], with ground truth actions from the phase 1 policy. We use student predicted motor commands to step the environment. We initialize the actor network with a copy from phase 1 to minimize the drift when we directly step the environment with student actions. However for predicted heading, the depth encoding network is not pre-trained. Directly using predicted heading as observation could result in catastrophic distribution drift leading to incorrect action labels from the teacher. To overcome this issue, we propose to use a mixture of teacher and student (MTS). Concretely, the heading command the student observes

$$obs_\theta = \begin{cases} \theta pred, & \text{if } |\theta pred - \hat{\mathbf{d}}_w| < 0.6 \\ \hat{\mathbf{d}}_w, & \text{otherwise} \end{cases}$$

where $\theta pred$ and $\hat{\mathbf{d}}_w$ are the desired yaw angle from prediction and oracle, respectively. $obs_\theta$ is the yaw angle the policy observes.

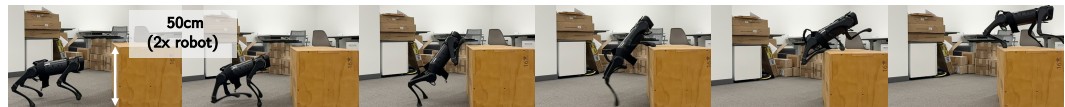

Figure 4: Key frames of our robot executing a very high jump (2x its height). We note the emergent foot placement, power generated through hind legs and climbing behavior from the front legs.

# 4   Results

## 4.1   Experimental Setup

We use the Unitree A1 robot with 12 joints and IssacGym simulator for training. When standing, height of the thigh joint is 26cm and body length is 40cm. For exteroception, we use the Intel RealSense D435 inside the head of the robot which captures images at $10 \pm 2$Hz. We run both depth backbone (10Hz) and the base policy (50Hz) on the Jetson NX and communicate via UDP. The depth server captures depth images, processes them and passes the latent and target direction to the base policy. We preprocess the image by cropping dead pixels from the left hand side and downsampling to $58 \times 87$. We enforce a constant depth latency of 0.08s to prevent jitter. Specifically, we record the time from receiving the depth image to the time before sending the latent as $t_p$. If $t_p < 0.08$, we pause the program for $0.08 - t_p$ before sending the latents. Similarly, proprioception latency is fixed at 0.016s. The deployable policy can be trained on a single 3090 GPU in less than 20 hours.

## 4.2   Emergent results

Our simple reward functions impose no priors and the robot is free to learn emergent behaviors that would be impossible to heuristically define. We illustrate three such examples in Fig. 4, 5, 6.

### 4.2.1   High jump

Our robot is able to jump on a gym box 0.5m high (Fig. 4) which is twice the height of its hip joint. For context, the human high jump record is 2.45m which is roughly 2.5 times as high as the human hip joint. This feat is only possible with highly optimized behavior.

In Fig. 4 we show a breakdown. As the robot approaches the obstacle the stride length reduces and the robot aligns its front feet and rear feet at the correct distance from the obstacle. Next, it kicks out its rear feet with high torque and velocity to propel itself upwards. Simultaneously, it extends its front feet to clear the top of the obstacle. As soon as the front feet touch the top of the obstacle, it uses them to pull itself up. Next, it tucks its rear legs close to the body so they are able to clear the object boundary and then finally shifts to a stable walking pose.

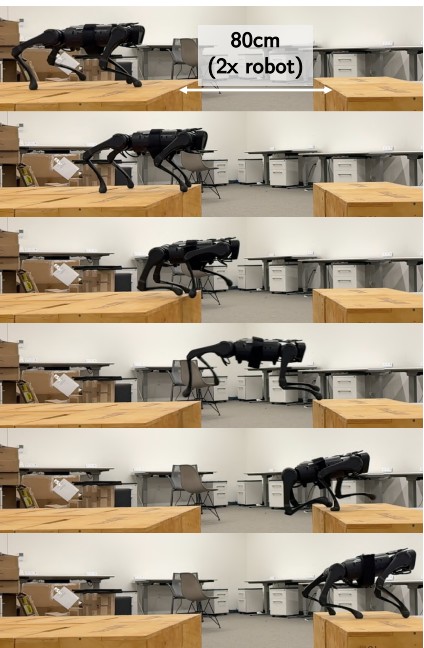

Figure 5: Keyframes from a long jump (2x robot length)

### 4.2.2   Long jump

Our robot is able to jump across a gap 0.8m wide (Fig. 5). This is twice the separation between its front and rear feet. To accomplish this, similar to the high jump case it lines up its front feet with the edge of the obstacle. Next, it also moves its rear feet close to the edge to maximize its jumping distance. Then it kicks back using its hind feet to propel itself forward and upward while extending the front ones to reach the other side. While jumping, it extends its hind feet to maximize the duration of force application. Once it is in midair it extends its front feet and moves the hind ones close to them such that they both land on the far side. After landing safely, it extends its front ones again to shift to a normal gait.

### 4.2.3   Handstand

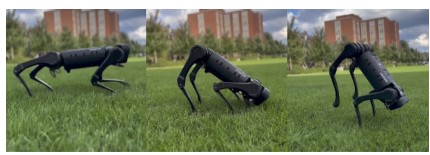

Figure 6: Transition from quadrupedal walking to bipedal walking.

Our robot can seamlessly transition between walking on four and its front two legs (Fig. 6). Bipedal walking in general is a much harder task than quadrupedal walking since a four-legged system is more inherently stable. For example, a quadruped in its canonical pose will remain standing and is a stable system, however a biped will topple unless small active adjustments are constantly made. Our robot learns to make these adjustments and is even able to do a handstand walk on soft deformable grass with gentle slopes (Fig. 6). To transition into a handstand, it first bends forwards and shifts its entire weight onto the front legs. Then, it kicks upward with its rear legs just the right amount to move into a vertical position. Once in the vertical position, it keeps its rear legs in a neutral pose and makes tiny adjustments to them to maintain balance. Due to the robustness of the handstand policy, our robot is able to descend stairs in a handstand pose without vision and stabilize against the sudden dips.

## 4.3 Comparison to Baselines

We propose two sets of baselines to experimentally verify different parts of our system. First, we test our reward design and overall pipeline (Tab. 2):

- **Noisy:** This simulates a system that uses an elevation map constructed by fusing depth and odometry. As in [2], we simulate sensor noise in the map and train the phase 1 policy. This tests if an end-to-end system is more performant over a modular one that relies on elevation mapping.

- **No inner product reward (NoInner):** This replaces the inner product reward Eq. 2 with velocity tracking in base frame used in [2].

- **No feet clearance penalty (NoClear)**: Removes the penalization for stepping near the edges defined in Eq. 3.

The second set of baselines is designed to test our distillation setup which involves BC for the direction prediction and dagger for actions (Tab. 3).

- **Both**: Student always observes predicted yaw angles.

- **Mask**: The yaw angle observations are masked with zeros in phase 2 and the student is trained via action supervision to learn turning behaviors with no specified direction command.

- **Oracle**: Student observes oracle yaw angles from the waypoints.

### 4.3.1 Simulation results

For each terrain—tilted ramps, steps, gaps and hurdles we create an obstacle course consisting of each arranged in increasing difficulty in series. We spawn 256 robots at the beginning of the course and record the mean x-displacement (MVD) before they fall and the average number of times per time-step a robot steps on an edge (mean edge violation - MEV). A larger value for the former while a lower for the latter is desirable since stepping on the edge is unstable in the real world.

We find that our method outperforms the baselines in terms of both metrics. The *NoInner*'s behavior on hurdle terrain is to walk around the obstacle instead of getting over it, so it has lowest edge violation metric. It struggles especially on step terrain because there is no way to get around the obstacle and still get to the next waypoint. All it learns is a colliding and retrying behavior where the robot first walk and use its feet to bounce back from the high step and walk forward again. *NoClear* achieves slightly higher performance but it places feet close to the obstacle edges which is unstable in the real world. *Noisy* is able to get some performance but has very large variance since it can rely on collisions with its legs to sense terrain geometry and overcome noise in the map. In addition, its feet clearance also helps it to achieve some performance with noisy measurements. We omit the *NoDir* baseline comparison in simulation since it is infeasible to provide human joystick commands and provide real-world comparisons instead.

Similarly, we compare against the distillation baselines in Tab. 3 averaged across all terrain. We find that ours is very close to the upper bound which receives oracle direction commands and it does much better in terms of x displacement as compared to *Mask* and *Both* since they fail to converge to low loss values since the data distribution used for imitation learning drifts significantly from the teacher.

| Terrain | Mean $X$-Displacement (MXD) $\uparrow$ | | | | Mean Edge Violation (MEV) $\downarrow$ | | | |
|---|---|---|---|---|---|---|---|---|
| | Ours | NoInner | NoClear | Noisy | Ours | NoInner | NoClear | Noisy |
| Hurdle | 0.99±0.05 | 0.90±0.12 | 1.00±0.03 | 0.78±0.26 | 0.04±0.21 | 0.03±0.16 | 0.12±0.38 | 0.31±0.58 |
| Step | 0.99±0.07 | 0.14±0.00 | 1.00±0.04 | 0.84±0.29 | 0.04±0.20 | 0.04±0.21 | 0.07±0.27 | 0.15±0.38 |
| Gap | 0.96±0.14 | 0.86±0.26 | 0.96±0.12 | 0.87±0.24 | 0.02±0.14 | 0.04±0.20 | 0.07±0.32 | 0.06±0.25 |
| Ramps | 1.00±0.04 | 0.92±0.24 | 1.00±0.03 | 0.79±0.31 | 0.01±0.11 | 0.02±0.13 | 0.04±0.19 | 0.14±0.41 |
| Total | 0.98±0.09 | 0.75±0.36 | 0.99±0.06 | 0.82±0.29 | 0.03±0.18 | 0.04±0.20 | 0.08±0.32 | 0.20±0.50 |

Table 2: We create a simulated obstacle course consisting of versions of each terrain arranged in increasing levels of difficulty and measure the average displacement along $x$ and the mean time until failure for 256 randomly spawned robots in 30s. We report the mean maximum number of waypoints reached normalized to [0, 1] indicating the policy's capability on different terrains, and the mean edge violation computed by taking the average of feet contact counts on edges.

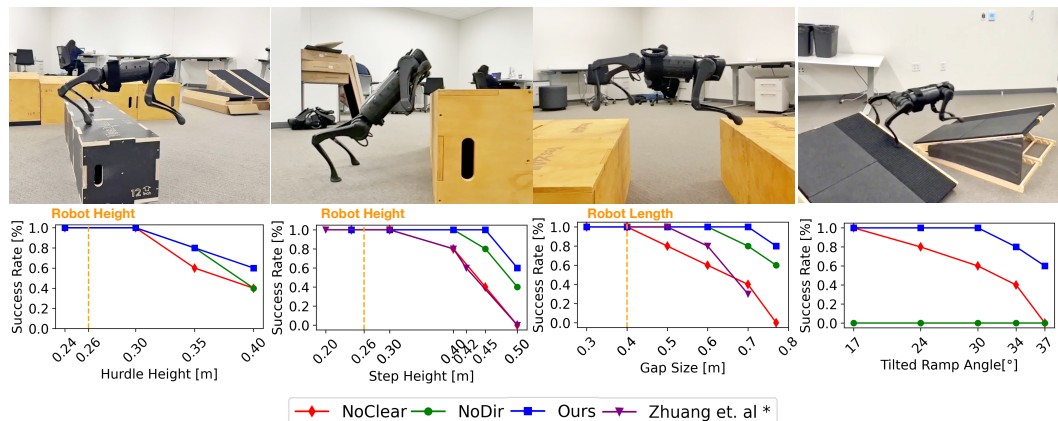

Figure 7: For each terrain, we run 5 trials and record the number of successes. We find that ours has 20-80% higher success rate on the most difficult instance of each terrain. NoDir is provided direction commands via a joystick controlled by a trained human operator. It sometimes succeeds on hurdles and gaps but fails when the human has to provide sudden direction changes which are out-of-distribution. It also fails on tilted ramps which require sudden direction changes hard for a human to do. NoClear is trained without feet edge penalty and therefore steps very close to the edge which is unstable and often falls. Starred is recent concurrent work [14].

### 4.3.2 Real-world results

We compare against NoClear and NoDir baselines in the real world. Each method is run for 5 trials on each terrain for each difficulty and the success rate is recorded (Fig. 7). For the NoDir baseline, directions commands are provided via joystick by a trained human operator. We find that ours has much higher success rate in all environments. NoDir fails on jumps and gaps when the operator has to make last-minute yaw adjustment to keep the robot perpendicular to the obstacle. These sudden adjustments are out-of-distribution for the policy and it cannot adapt fast enough, causing it to fail. We find that NoDir is especially bad

| | MXD $\uparrow$ | MEV $\downarrow$ |
|---|---|---|
| Both | 0.12±0.07 | 0.26±0.57 |
| Mask | 0.05±0.07 | 0.00±0.00 |
| Ours | 0.92±0.19 | 0.09±0.33 |
| Oracle | 0.94±0.19 | 0.10±0.32 |

Table 3: Ours reaches almost the same performance as oracle yaw angles as inputs. Both and Mask work poorly because the noisy yaw angle leads to large drift.

on tilted ramps since this requires quick changes in direction which are tricky for a human to do. The NoClear baseline without clearance reward $r_{clearance}$ tends to place feet very close to the gap or cliff edge since this minimizes energy usage. We find that this is unstable behavior and the robot sometimes misses the edge and falls.

For the handstand walking policy, we train it without exteroception. Despite this, we find it has strong robustness not only on different types of terrain (indoor and outdoor), but can also walk down the stairs using proprioception alone.

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
