# OpenReview forum: "Extreme Parkour with Legged Robots"
_robot-learning.org/CoRL/2023/Workshop/TGR — CoRL 2023 Workshop TGR Oral_

### Official Review · Reviewer_BWvD · 2023-10-19
**Strong accept**

**Rating:** 10
**Confidence:** 4

**Review:**

Amazing sim-to-real transfer results with fascinating leg motion during dynamic parkour. This work brings legged robot control to a new level. congrats

---

### Decision · Program_Chairs · 2023-10-20

**Decision:**

Accept (Oral)

**Comment:**

Great paper! Learning diverse and complex locomotion skills is an important step towards generalist robot learning.